# Efficacy Studies against PCV-2 of a New Trivalent Vaccine including PCV-2a and PCV-2b Genotypes and *Mycoplasma hyopneumoniae* When Administered at 3 Weeks of Age

**DOI:** 10.3390/vaccines10122108

**Published:** 2022-12-09

**Authors:** Patricia Pleguezuelos, Marina Sibila, Carla Ramírez, Rosa López-Jiménez, Diego Pérez, Eva Huerta, Anna Maria Llorens, Mónica Pérez, Florencia Correa-Fiz, José Carlos Mancera Gracia, Lucas P. Taylor, Jennifer Smith, Meggan Bandrick, Stasia Borowski, Gillian Saunders, Joaquim Segalés, Sergio López-Soria, Maria Fort, Mónica Balasch

**Affiliations:** 1Unitat Mixta d’Investigació IRTA-UAB en Sanitat Animal, Centre de Recerca en Sanitat Animal (CReSA), Campus de la Universitat Autònoma de Barcelona (UAB), Bellaterra, 08193 Barcelona, Spain; 2IRTA, Programa de Sanitat Animal, Centre de Recerca en Sanitat Animal (CReSA), Campus de la Universitat Autònoma de Barcelona (UAB), Bellaterra, 08193 Barcelona, Spain; 3OIE Collaborating Centre for the Research and Control of Emerging and Re-Emerging Swine Diseases in Europe (IRTA-CReSA), Bellaterra, 08193 Barcelona, Spain; 4Zoetis Belgium S.A., 20 Mercuriusstraat, 1930 Zaventem, Belgium; 5Zoetis Inc., 333 Portage Street 300-504SW, Kalamazoo, MI 49007, USA; 6Departament de Sanitat i Anatomia Animals, Facultat de Veterinària, Universitat Autònoma de Barcelona, Bellaterra, 08193 Barcelona, Spain; 7Zoetis Manufacturing & Research Spain S.L., Ctra Camprodon s/n Finca “La Riba”, Vall de Bianya, 17813 Girona, Spain

**Keywords:** porcine circovirus 2, postweaning multisystemic wasting disease, porcine respiratory disease complex, trivalent vaccine, efficacy, single dose

## Abstract

This study aimed to evaluate the efficacy of a new trivalent vaccine containing inactivated Porcine Circovirus 1-2a and 1-2b chimeras and a *Mycoplasma hyopneumoniae* bacterin administered to pigs around 3 weeks of age. This trivalent vaccine has already been proved as efficacious in a split-dose regimen but has not been tested in a single-dose scenario. For this purpose, a total of four studies including two pre-clinical and two clinical studies were performed. Globally, a significant reduction in PCV-2 viraemia and faecal excretion was detected in vaccinated pigs compared to non-vaccinated animals, as well as lower histopathological lymphoid lesion plus PCV-2 immunohistochemistry scorings, and incidence of PCV-2-subclinical infection. Moreover, in field trial B, a significant increase in body weight and in average daily weight gain were detected in vaccinated animals compared to the non-vaccinated ones. Circulation of PCV-2b in field trial A and PCV-2a plus PCV-2d in field trial B was confirmed by virus sequencing. Hence, the efficacy of this new trivalent vaccine against a natural PCV-2a, PCV-2b or PCV-2d challenge was demonstrated in terms of reduction of histopathological lymphoid lesions and PCV-2 detection in tissues, serum and faeces, as well as improvement of production parameters.

## 1. Introduction

Porcine circovirus 2 (PCV-2) is a single-stranded DNA (ssDNA) ubiquitous virus in swine and the causative agent of the so-called Porcine Circovirus Diseases (PCVD), which may be manifested as subclinical or clinical infections [1]. The subclinical infection (PCV-2-SI) is the most common form of PCV-2 infection outcome [1], causing a loss of average daily weight gain (ADWG) without evident clinical signs and, consequently, significant economic losses [2,3]. PCV-2 systemic disease (PCV-2-SD) is characterised by loss of weight, digestive signs, paleness of skin and dyspnoea in pigs mainly between six and eleven weeks of age. This is the most severe outcome causing significant economic impact in the swine industry worldwide [4,5].

As other ssDNA, PCV-2 has a high mutation rate, being around 10^3^–10^4^ substitutions/site/year [6]. Additionally, PCV-2 can also evolve by means of recombination [7,8,9]. These two factors result in genetic changes and a wide diversity within PCV-2 [6,10]. In fact, nowadays, nine genotypes of PCV-2 are recognised (a–i) based on ORF2 gene sequencing [8,11], three of them (a, b and d) being the most frequently associated with clinical PCV2-SD [12]. PCV-2a was the most prevalent genotype from 1996 to the early 2000s. Then, a genotype shift occurred coinciding with the change of PCV-2-SD prevalence from sporadic to epidemic, becoming PCV-2b predominant [13,14,15,16,17]. Later, PCV-2d increased gradually worldwide from 2012, becoming a second global genotype shift from PCV-2b to PCV-2d [7,18,19,20].

Vaccination is the primary tool to reduce PCV-2 infection. In fact, PCV-2a vaccines have successfully decreased the prevalence and severity of PCV-2 viral infections [21]. Nowadays, PCV-2 vaccines in Europe are based on the PCV-2a genotype or a combination of PCV-2a and PCV-2b genotypes [22,23,24]. Interestingly, some studies showed a closer genetic and antigenic relation between PCV-2b and PCV-2d sequences compared to PCV-2a and PCV-2d sequences [20,23]. Hence, a bivalent vaccine containing PCV-2a and PCV-2b genotypes may be a relevant option to face against the several PCV-2 genotypes that are circulating under field conditions [25,26]. In addition to PCV-2, *Mycoplasma hyopneumoniae* (*M. hyopneumoniae*) is also an important pathogen that usually circulates in pigs during the postweaning period. *M. hyopneumoniae* is the main causative agent of enzootic pneumonia (EP) and one of the main contributors of porcine respiratory disease complex (PRDC), a multimicrobial and multifactorial condition in which different bacterial and viral agents are involved, including PCV-2 [27,28,29,30]. Therefore, combination of PCV-2 and *M. hyopneumoniae* in one ready-to-use vaccine is a relevant option to reduce the management of the animals and, consequently, reduce the stress and the management of associated costs [31,32,33]. In fact, the number of combined vaccines including PCV-2 and *M. hyopneumoniae* has increased in the last few years, with seven PCV-2/*M hyopneumoniae* combined vaccines in the market nowadays.

The current work aimed to elucidate the efficacy against PCV-2a or PCV-2b experimentally or naturally infected pigs of a new trivalent vaccine containing inactivated porcine circovirus 1 (PCV-1)/PCV-2a and PCV-1/PCV-2b chimeras (cPCV-1/2a, cPCV-1/2b) as well as *M. hyopneumoniae* (CircoMax Myco^®^) administered in one dose at three weeks of age. Different PCV-2 vaccination regimens are already studied and commercialised. In fact, this chimeric vaccine has already shown efficacy in pre-clinical and clinical trials applied in a split-dose immunisation regime at three days of age and three weeks later [26]. This latter regime is especially interesting in those farms where PCV-2 infection is detected at early stages of life (lactating or early nursery periods) or when herd immunity is poor. However, in those farms with PCV-2 infection detected later, a single-dose vaccination should be enough to prevent or reduce a PCV-2 disease outbreak. Moreover, further advantages of a single dose regime include reduction of management and vaccination costs.

## 2. Materials and Methods

A total of two pre-clinical and two clinical studies were conducted. Each of these assays were independently performed and evaluated by different regulatory agencies (pre-clinical studies by the U.S. Department of Agriculture (USDA) regulatory agency and clinical ones by European Medicines Agency (EMA) regulatory agency). For this reason, requirements for efficacy parameters were different and some laboratory analyses were conducted with different methodological approaches and using distinct interpretation criteria. Efficacy variables considered in pre-clinical studies were PCV-2 antibody levels, PCV-2 viraemia and faecal shedding and PCV-2 detection in lymphoid tissues and microscopic lymphoid lesions. In case of clinical trials, the primary efficacy variable was the PCV-2 viraemia as determined by real time quantitative PCR (qPCR) and the following variables were considered as secondary variables: PCV-2 antibody levels, body weight, ADWG, the correlation between PCV-2 antibody levels before vaccination and ADWG, mortality, PCV-2 faecal shedding, PCV-2 detection in lymphoid tissues, microscopic lymphoid lesions and PCV-2-SD or PCV-2-SI diagnoses.

### 2.1. Pre-Clinical Studies

Two experimental studies were carried out including vaccinated and non-vaccinated pigs challenged with PCV-2a or PCV-2b. Both were approved by the corresponding institutional animal care and use committees (IACUC) from Zoetis (KZ-3226e-2017-06-tkh for the PCV2a study and PJ017 for the PCV2b study).

#### 2.1.1. PCV-2a Challenge Study

The study design is described in Table 1. A total of 25 clinically healthy animals with maternally derived antibodies (MDA) complied with the inclusion criteria and they were enrolled in the non-vaccinated group, and 23 were in the vaccinated one. Vaccinated animals were treated with one dose of the Investigational Veterinary Product (IVP) containing Porcine Circovirus Vaccine, cPCV-2a and cPCV-2b killed viruses, and *M. hyopneumoniae* bacterin adjuvanted with 10% SP Oil (equivalent to Fostera Gold^®^ and CircoMax Myco^®^) in a single dose of 2 mL by the intramuscular (IM) route. The non-vaccinated group received a placebo containing a *M. hyopneumoniae* bacterin adjuvanted with 10% SP Oil in the same volume and by the same inoculation administration.

Following vaccination and until challenge, blood was collected from pigs at weekly intervals and tested by ELISA for PCV-2 antibody detection. All pigs were challenged with a PCV-2a field strain (4 mL dose; 2 mL IM and 2 mL intranasally (IN)) five weeks post-vaccination (SD35), when the S/P ratio of mean MDA titre was ≤0.2 in the non-vaccinated group, ensuring maximal susceptibility to viral infection. The PCV-2a field strain (isolate 40895; GenBank accession number AF2640) diluted 1:2 in Optimem (Gibco) was used for PCV-2a challenge. The diluted stock material had a titre of 10^5.6^ TCID_50_/mL.

Following challenge, blood and faecal swabs were collected two times every week to determine PCV-2 viraemia and shedding by qPCR. PCV-2 antibodies were also measured by ELISA at weekly intervals post-challenge. Pigs were euthanised and necropsied at 21 days post challenge (SD56). Lymphoid tissues (tracheobronchial, mesenteric and superficial inguinal lymph nodes and tonsil) were collected, and they were processed for histopathology and PCV-2 immunohistochemistry (IHC).

#### 2.1.2. PCV-2b Challenge Study

The study design is described in Table 1, following the same schedule indicated for the PCV-2a challenge experiment. A total of 19 clinically healthy animals with MDA in the non-vaccinated group, and 16 in the vaccinated one, complied with the inclusion criteria and were enrolled in the study. When non-vaccinated pigs had a mean MDA titre corresponding to S/P ≤0.2 (approx. SD41), a PCV-2b challenge (4 mL dose; 2 mL IM and 2 mL IN) was performed. The PCV-2b strain (isolate FD07; GenBank accession number GU799576) diluted 1:2 in Optimem (Gibco) was used for the challenge with a final titre of 10^5.3^ TCID_50_/mL. All pigs were euthanised at SD61-62 and the same procedures and sampling were performed during necropsy as indicated in the PCV-2a challenge study.

### 2.2. Field Studies

#### 2.2.1. Farm Selection

A total of two field trials were conducted in two different Spanish commercial farms. Criteria for farm selection were the existence of problems with PCVD or history of PCVD in the last two-and-a-half years.

Farm A was a two-site commercial farm (breeding and gestation plus nursery) with 2660 sows and a weekly farrowing batch system. Piglet weaning was carried out around four weeks of age. The sow farm was seropositive against *M. hyopneumoniae*, Porcine reproductive and respiratory syndrome virus (PRRSV) and seronegative to Aujeszky’s disease virus (ADV). Gilts and sows were crossbred (Duroc x Landrace). The sow and gilt vaccination farm programme included PRRSV, Porcine parvovirus, *Erysipelothrix rhusiopathiae,* Swine influenza virus (SIV), *Actinobacillus pleuropneumoniae* and PCV-2 (the piglets at weaning, the gilts at 6 months of age and the sows post-partum) immunisations. At fattening facilities, pigs were vaccinated twice against ADV.

Farm B was a farrow-to-finish commercial farm with 10,500 sows with a weekly farrowing batch system. Piglet weaning was conducted at approximately 25 days of age. The sow farm was seropositive against *M. hyopneumoniae* and PRRSV and seronegative to ADV. Gilts and sows were of Pietrain breed. The sow and gilt vaccination farm programme included immunisation against PRRSV, SIV, Porcine parvovirus, *Erysipelothrix rhusiopathiae*, *Escherichia coli, Clostridium perfringens type C*, atrophic rhinitis, ADV, *M. hyopneumoniae* and PCV-2 (at 3 and 6 weeks of age). Gilts were also vaccinated against PCV-2 at two-and-a-half, six and seven months of age. Piglets were vaccinated against PRRSV before weaning and against ADV, PRRSV and SIV at fattening.

#### 2.2.2. Study Design

The design of these field studies was blinded, randomised and controlled. A total of 4076 male and female pigs (2037 vaccinated and 2039 non-vaccinated) were enrolled in two trials: A and B (Table 2).

The sample size used for each variable was calculated by a biometrician using data from field safety and efficacy studies previously performed [34].

The number of animals in each batch was determined by the number of clinically healthy pigs available on the particular week of study initiation. Thus, field trial A required recruitment of pigs from three different batches, while for the field trial B, one batch was enough. Selection of pigs included in the study and distribution (blocked by gender) in vaccinated and non-vaccinated groups were conducted between SD-3 and SD0, SD0 being the vaccination day.

Study animals were clinically observed daily throughout the study. A single vaccination was performed at three weeks of age approximately with 2 mL of a trivalent vaccine containing inactivated cPCV-1/2a, cPCV-1/2b chimeras and *M. hyopneumoniae* bacterin (CircoMax Myco^®^, Zoetis Inc., Lincoln, NE, USA) by IM route in the neck. Non-vaccinated pigs received 2 mL of phosphate buffer saline (PBS). Pigs from each treatment group were housed comingled in the maternity and nursery phase, but males and females were separated by pen at fattening (in each pen there were vaccinated and non-vaccinated animals from the same gender).

Blood samples and faecal swabs from piglets were collected at 7, 11, 16, 20 and 25 weeks of age approximately. Blood samples were also collected at three weeks of age (just before vaccination). Sera samples were analysed by a validated in-house PCV-2 antibody ELISA and by a qPCR assay and faecal swabs were analysed by qPCR. Moreover, the body weight was registered before vaccination, at 16 weeks of age and before slaughter for 400 animals approximately for each treatment group (a minimum of 328 animals and a maximum of 438 as indicated in Appendix A). Animals weighed during the study were not the same at each timepoint due to deviations occurring during the study (animal deaths or animals not found at the weighing moment) as indicated in Appendix A. Thus, extra animals from the same treatment group were selected for weighing when any animal selected for this action was missing.

Dead animals or pigs euthanised for welfare reasons from weaning until the slaughterhouse were examined post-mortem to determine the cause of death. Tissue samples collected at each necropsy (tracheobronchial, mesenteric and superficial inguinal lymph nodes and tonsil) were processed for histopathology and PCV-2 IHC for PCVD diagnosis performed by a pathologist blinded to the treatment status. Moderate and severe histological lesions together with a moderate or high amount of PCV-2 antigen in lymphoid tissues were diagnosed as PCV-2-SD [1]. When a PCV-2-SD diagnosis was confirmed in the studied herd, 60 animals (30 per treatment group) were selected and necropsied to obtain the above-mentioned lymphoid tissue samples. These samples were analysed by histopathology and PCV-2 IHC.

The Cap gene (ORF2) from 20 serum samples with the highest PCV-2 viral load (6.3–8.3 log^10^ DNA copies/mL) belonging to non-vaccinated groups was sequenced to determine the PCV-2 genotype/s circulating in the farms.

Clinical studies were approved by the Olot Animal Welfare Committee (ID PJ023) and performed according to Guidelines on Good Clinical Practices.

#### 2.2.3. PCV-2 Genotyping

Total DNA was extracted from serum samples, amplified and sequenced. Then, a phylogenetic analysis was performed, and a phylogenetic tree was edited. All these procedures were performed as described in Pleguezuelos et al.’s study [26].

### 2.3. Laboratory Methods of Pre-Clinical and Field Studies

#### 2.3.1. DNA Extraction and PCV-2 qPCR

DNA from pre-clinical serum and faecal samples were extracted and qPCR-analysed with a non-commercial in-house qPCR as indicated in Mancera-Gracia et al. [6]. No threshold was applied; therefore, all detected values were reported as positive. In the case of clinical studies, serum and faecal samples were extracted and qPCR-analysed with a commercial kit LSI VetMAXTM Porcine Circovirus Type 2-Quantification Applied Biosystems, Lisseu, France) [26]. The limit of detection (LOD) of the technique in serum samples was 4 × 10^3^ DNA copies/mL and in faecal swabs it was 1 × 10^4^ DNA copies/mL. The limit of quantification (LOQ) in serum samples and faecal swabs was 1 × 10^4^ DNA copies/mL. Log_10_ transformation of qPCR results was conducted, and the results were interpreted as follows:-Negative results or values below LOD were given a value equal to half of the LOD (log_10_ 3.3 copies/mL for serum samples and log_10_ 3.7 copies/mL for faecal swabs).-Values between LOD and LOQ were considered positive and were given a value equal to LOQ (log_10_ 4.0 for serum and faecal swabs).-Values over LOQ were considered positive and were given the log_10_ qPCR result obtained.

#### 2.3.2. PCV-2 Serology

Pre-clinical and clinical PCV-2 antibodies were detected using a validated in-house PCV-2 antibody ELISA [26]. Sera samples with sample/positive control (S/P) ratio (OD sample–OD negative control/OD positive control–OD negative control) values ≥ 0.5 were considered positive.

#### 2.3.3. Histopathology and PCV-2 IHC

Lymphoid samples collected at necropsy (tracheobronchial lymph node, mesenteric lymph node, superficial inguinal lymph node and tonsil) were fixed by immersion in 10% buffered formalin and examined for lesions compatible with PCV-2, including lymphocyte depletion (LD) and histiocytic replacement (HR). Moreover, another section was cut for PCV-2 antigen detection by IHC [5]. LD, HR and the amount of PCV-2 antigen were scored from 0 (no lesions/no staining) to 3 (severe lesions/widespread antigen distribution) for each lymphoid tissue collected.

In field trials, dead or euthanised pigs from weaning age were classified as PCV-2-SD or PCV-2-SI, following the diagnostic criteria indicated below:

Presence of at least one of the following clinical signs: wasting, weight loss, paleness of the skin, dyspnoea, diarrhoea, jaundice and/or inguinal superficial lymphadenopathy (only applicable to PCV-2-SD cases).

LD and/or HR of lymphoid tissues (PCV-2-SI: LD and HR ≤ 1; PCV-2-SD: LD and HR > 1).

PCV-2 detection in lymphoid tissues (PCV-2-SI: IHC ≤ 1; PCV-2-SD: IHC > 1).

### 2.4. Statistical Analyses

Pre-clinical and field trials statistical analyses were carried out using the software SAS/STAT (User’s Version 9.4, or higher, SAS Institute, Cary, NC, USA). A logarithm transformation was applied to the data before statistical analyses were conducted when needed. Comparisons were performed between treatment groups (vaccinated vs. non-vaccinated) from each trial.

A general linear repeated measures mixed model was performed to analyse the following variables from pre-clinical and field studies in each study: sera and faecal qPCR results, ELISA S/P values and body weight.

Linear functions of the least squares mean for body weights were used to calculate estimates of the ADWG for each period. Moreover, a Pearson Correlation Coefficient was also calculated to evaluate the correlation between PCV-2 antibodies before vaccination and the ADWG during the whole study. A generalised linear mixed model was performed to analyse the following variables from pre-clinical and field studies in each study: ever positive (detected positive on at least one sampling point) for viraemia/shedding, mortality, LD, HR and IHC results separately, and diagnosis of PCV-2-SD or PCV-SI. When the mixed model did not converge, Fisher’s Exact test was used for analysis. Additionally, MDA’s effect on seroconversion of vaccinated piglets from field trials was evaluated by calculating a Pearson Correlation Coefficient between PCV-2 antibodies before vaccination and the increase of PCV-2 antibodies at seven weeks of age (Delta value).

The significance level (α) was set at *p* ≤ 0.05 for all statistical analyses.

## 3. Results

### 3.1. Pre-Clinical Studies

#### 3.1.1. PCV-2a Challenge

##### Clinical Evaluation

No clinical signs or mortality due to treatment administration or challenge were recorded in any studied group.

One animal from the non-vaccinated group was found dead at SD31 due to oedema disease. Additionally, one animal from the vaccinated group showed a left hind leg lameness at SD5. After several days with an anti-inflammatory treatment, no response was observed, and it was removed from study.

##### PCV-2 Antibody Detection

Mean ELISA S/P ratios results are represented in Figure 1A. The least squares mean of the S/P ratio decreased from SD0 to SD21 in both experimental groups. After challenge, S/P ratios in the vaccinated group were significantly higher (*p* ≤ 0.05) than the non-vaccinated group at any timepoint tested post-challenge.

##### PCV-2 Viraemia and Faecal Shedding

All pigs included in the study were negative for PCV-2 qPCR before the challenge (SD39). PCV-2 viraemia was initially detected seven days post-challenge. From SD42 until SD49, viral load in serum in the vaccinated group was significantly lower (*p* ≤ 0.01) than in the non-vaccinated one (Figure 1B). Moreover, the percentage of ever viraemic pigs was significantly lower (*p* ≤ 0.01) in the vaccinated group compared to the non-vaccinated one (Table 3).

Faecal shedding was in all cases with low viral load and was initially detected in the non-vaccinated group four days after challenge and continued through the end of study (Figure 1C). Additionally, from SD46 and until the end of the study (SD56), faecal shedding was significantly lower (*p* ≤ 0.01) in the vaccinated group compared to the non-vaccinated one. The percentage of pigs ever positive for faecal shedding was significantly lower (*p* ≤ 0.01) in the vaccinated group than in the non-vaccinated group (Table 3).

##### PCV-2 Detection in Lymphoid Tissues and Microscopic Lymphoid Lesions

PCV-2 virus antigen was detected only in very few animals from the non-vaccinated group in lymphoid tissues by IHC and there were no significant differences among treatment groups. Additionally, the percentage of pigs with LD and HR were not significantly different among treatment groups. The percentage of non-vaccinated pigs with lesions was very low (Table 4). HR and LD were considered in a combined analysis, as HR is combined with LD in cases of PCV-2-SD, but there were no significant differences among treatment groups (Table 4).

#### 3.1.2. PCV-2b Challenge

##### Clinical Evaluation

No clinical signs or mortality were recorded in any studied group due to treatment or after challenge.

##### PCV-2 Antibody Detection

The mean PCV-2 ELISA S/P ratio results obtained during the study are represented in Figure 2A.

All the pigs included in the MDA-positive groups had moderate levels of PCV-2 antibodies (ELISA S/P ratios ranging from 0.5 to 1.3). After challenge, a boost in PCV-2 antibody values was observed in the vaccinated group starting at SD48 (seven days post-challenge). The levels of PCV-2 antibodies detected in vaccinated pigs were significantly (*p* ≤ 0.01) higher than those from non-vaccinated ones at all timepoints after challenge (SD48, SD55 and SD61/62).

##### PCV-2 Viraemia and Faecal Shedding

The viral load detected in serum was significantly lower (*p* ≤ 0.05) in the vaccinated group compared to the non-vaccinated one in the whole post-challenge period (from SD48 to SD61/62), except for SD45 (Figure 2B). Additionally, the amount of PCV-2 detected in faecal swabs was significantly lower (*p* ≤ 0.01) in the vaccinated group compared to non-vaccinated one at SD59 and SD61/62 (Figure 2C). No significant differences were detected in the percentage of pigs ever positive in serum and faeces among groups (Table 3).

##### PCV-2 Detection in Lymphoid Tissues and Microscopic Lymphoid Lesions

The percentage of pigs with positive IHC scores in any of the lymphoid tissues evaluated was significantly higher (*p* ≤ 0.05) in non-vaccinated animals when compared to the vaccinated ones. No significant differences were detected between vaccinated and non-vaccinated pigs in the percentage of animals with PCV-2-associated lesions in any of the lymphoid tissues evaluated (Table 4).

### 3.2. Field Studies

#### 3.2.1. Clinical Evaluation

Body weight results and the ADWG are represented in Table 5. No significant differences in terms of body weight and ADWG were observed among vaccinated and non-vaccinated groups of field study A at any time. In field study B, a significantly higher (*p* ≤ 0.05) body weight was observed in the vaccinated group at 16 and 24–27 weeks of age (1–5 days before the slaughterhouse) compared to the non-vaccinated one. Additionally, in study B, ADWG from vaccinated animals was significantly higher (*p* ≤ 0.05) in the three periods (from three weeks of age to 16 weeks of age, from 16 weeks of age to 24–27 weeks of age and from three weeks of age to 24–27 weeks of age) than non-vaccinated group.

No significant differences were detected in mortality between treatment groups in either field trial.

No significant correlation was observed between PCV-2 antibody levels before vaccination and ADWG was detected in vaccinated groups of either field trial, indicating that ADWG of vaccinated pigs was independent of ELISA S/P titres at vaccination.

#### 3.2.2. PCV-2 Antibody Detection

No significant differences between treatment groups in mean PCV-2 S/P ratios before the time of vaccine/placebo administration were found in any of the field trials (Figure 3A and Figure 4A).

In field trial A, piglets from the vaccinated group had significantly higher (*p* ≤ 0.05) mean PCV-2 antibodies from 7 to 16 weeks of age compared to those of the non-vaccinated one (Figure 3A).

In field trial B, piglets from the vaccinated group had significantly higher (*p* ≤ 0.05) mean S/P values at 11 and 16 weeks of age. In contrast, a significantly lower (*p* ≤ 0.05) mean PCV-2 S/P ratio was detected at 25 weeks of age compared to the non-vaccinated group (Figure 4A).

The correlations between the PCV-2 ELISA S/P values of vaccinated animals before immunisation and their increase at seven weeks of age (Delta value) are represented in Figure 5A,B. A significantly negative (*p* ≤ 0.05) correlation between IgG ELISA S/P values and PCV-2 antibody levels at seven weeks of age was detected in vaccinated groups from both field studies, indicating that the higher the PCV-2 S/P of the mother before vaccination, the lower the increase in PCV-2 S/P values observed at seven weeks of age.

No significant correlation was obtained for the non-vaccinated groups in both field trials (data not shown).

#### 3.2.3. PCV-2 Viraemia

All tested pigs (*n* = 204) were PCV-2 qPCR-negative before vaccination.

A significantly lower (*p* ≤ 0.05) PCV-2 load and percentage of viraemic pigs was observed in vaccinated pigs from both field trials from 16 to 25 weeks of age compared to the non-vaccinated groups (Figure 3B and Figure 4B and Table 6).

In field trial A, the percentage of positive pigs peaked at 20 weeks of age in the non-vaccinated group (36/47 (76.6%)) and at 16 weeks of age in the vaccinated one (13/45 (28.9%)). The peak of viraemia (maximum viral load in serum) was observed at 16 weeks of age for both groups.

In field trial B, the percentage of positive pigs increased to a maximum of 100% (61/61) at 16 weeks of age in the non-vaccinated group. In the vaccinated group, it was obtained at seven weeks of age (28/48 (58.3%)) and decreased afterwards. The peak of viraemia was observed at 7 and at 16 weeks of age in the vaccinated and the non-vaccinated groups, respectively.

The percentage of pigs ever viraemic (detected positive at least at one sampling point) of both field trials was also significantly lower (*p* ≤ 0.05) in the vaccinated group compared to the non-vaccinated one (Table 6).

#### 3.2.4. PCV-2 Faecal Shedding

PCV-2 faecal shedding results from field trial A and B are summarised in Figure 3C and Figure 4C, respectively. In field trial A, statistically significant lower (*p* ≤ 0.05) PCV-2 faecal shedding was observed in vaccinated pigs at 25 weeks of age compared to non-vaccinated pigs. In field trial B, a statistically lower (*p* ≤ 0.05) PCV-2 load in faecal swabs was also detected in vaccinated pigs from 16 to 25 weeks of age than in non-vaccinated ones.

In field trial A, the peak of faecal shedding (maximum viral load in faeces) was observed at 20 weeks of age for both groups. In case of field trial B, peak faecal shedding was observed at 16 weeks of age for both groups.

Regarding the percentage of positive faecal swabs detected at least in one sampling point, no statistical differences were detected in any of the two studies between the vaccinated pigs (45/46 (97.8%) and 63/63 (100.0%) from field trials A and B, respectively) and the non-vaccinated ones (45/45 (100.0%) and 61/61 (100.0%) from field trials A and B, respectively).

#### 3.2.5. PCV-2 Genotyping

To determine the main PCV-2 genotype/s circulating in the farms during the study periods, a total of 20 PCV-2 qPCR-positive samples with the highest viral load (6.3–8.3 log_10_ DNA copies/mL), 10 for each field trial and belonging to non-vaccinated groups, were sequenced. A phylogenetic tree relating the ORF2 sequences obtained in these studies together with reference strains was built to determine the predominant genotypes present (Appendix A). In field trial A, genotype PCV-2b was found in nine out of ten serum samples. One serum sample failed to be sequenced. In field trial B, genotype PCV-2a was found in two serum samples, PCV-2d in four samples and no sequence was obtained from the remaining four sera.

#### 3.2.6. Histopathology and PCV-2 IHC

Histopathology and IHC results of the field trials are summarised in Table 7.

Non-vaccinated animals from both field studies showed a significantly higher (*p* ≤ 0.05) HR and positive PCV-2 IHC compared to vaccinated ones. Moreover, in field trial A, a significantly higher (*p* ≤ 0.05) incidence of PCV-2-associated lymphoid lesions (HR and LD together) was detected in non-vaccinated pigs than in vaccinated ones.

The number of cases diagnosed as PCVD-SD was 0.9% (1/116) and 2.4% (6/245) in the non-vaccinated groups from field trials A and B, respectively. In vaccinated groups, 0.0% (0/111) and 0.5% (1/218) of PCVD-SD cases were diagnosed in field trial A and B, respectively. Additionally, the number of PCVD-SI cases in non-vaccinated groups showed statistically higher (*p* ≤ 0.05) values (field trial A: 21 out of 116 (18.3%) and field trial B: 26 out of 245 (10.9%)) compared to those in vaccinated animals (field trial A: 4 out of 111 (3.6%) and field trial B: four out of 218 (1.8%)).

## 4. Discussion

PCVDs are causing great economical losses to the swine industry [1]. Vaccination of piglets against PCV-2 is the main control method to prevent PCVD in swine farms worldwide [35]. In general, combined vaccination of PCV-2 and *M. hyopneumoniae* around three weeks of age is one of the main strategies to reduce the impact of these two diseases [31,32,33].

PCV-2 vaccine benefits have been reported in terms of reduction in mortality [34], PCV-2 viraemia and lymphoid lesions [36,37], the frequency of co-infections and also the improvement of the ADWG [36,37,38,39,40] in PCV-2-SD scenarios. Moreover, an improvement of ADWG, percentage of runts, body condition and carcass weight has been also detected in the case of PCV-2-SI [3]. Interestingly, most PCV-2 vaccines in the market are based on the PCV-2a genotype. This is because the high degree of cross-protection between the major circulating genotypes worldwide (PCV-2b and PCV-2d) [22,41,42,43,44,45]. However, PCV-2 vaccines do not eliminate virus replication and transmission, and it has been speculated that broader-spectrum genotype-based vaccines may help in controlling better the infection under field conditions [21].

Hence, the aim of the present study was to evaluate the efficacy of a new trivalent vaccine containing inactivated cPCV-2a, cPCV-2b and *M. hyopneumoniae*, administered in piglets around three weeks of age. The efficacy of this vaccine has been previously demonstrated in a regime of split-dose immunisation at three days and three weeks of age [26], but it was important to ascertain its efficacy in a single-shot regime at the most common timing of vaccination against those two pathogens (around weaning). Both vaccination regimens for piglets are interesting for the swine industry and the selection of one or the other should depend on the dynamics of PCV-2 infection detected in the farm and the levels of herd immunity. A split-dose regime can help prevent early PCV-2 infections and provide solid immunity earlier in life. In the case of farms with late PCV-2 infection of piglets and significant herd immunity, a one-dose regime should be enough to counteract the detrimental effects of PCV-2 infection. In addition, one single short regime adds practical advantages such as a reduction of pig handling and stress, as well as economic ones such as less need for labour and human resources.

To accomplish the mentioned objective, four studies were carried out, including two pre-clinical studies performed in the USA and two clinical ones in the EU. Different regulatory agencies evaluated these studies; therefore, the requirements of each of the agencies were also different. Due to that, data of the qPCR were expressed differently in pre-clinical and clinical studies. However, the use of different qPCR techniques did not interfere in the analysis of the variables nor in the global efficacy assessment of the vaccine. Additionally, the interpretation of the results was not altered since the comparison between pre-clinical and clinical studies was not the goal of the present work.

Improvement of clinical variables such as signs compatible with PCVDs, body weight evolution, ADWG or mortality are usual claims of PCV-2 vaccines. However, these differences are unlikely to be detected under experimental settings with a limited number of animals and the fact that PCV-2 infection outcome is usually subclinical. Therefore, these claims are mostly demonstrated under field conditions, by means of large trials. In the present case, a significantly greater body weight at 16 and 24–27 weeks of age (one to five days before going to the slaughterhouse) and higher ADWG at the three periods (3–16 weeks of age, 16 to slaughter, and three weeks of age to slaughter) were observed in vaccinated pigs compared to non-vaccinated ones in field trial B. These differences in body weight were not statistically significant in field trial A. However, they showed a remarkable tendency for improvement of approximately 0.8 kg live weight at 16 weeks of age and 1.7 kg at slaughter, being an interesting improvement from an economical perspective [46]. These results are similar to those of several studies where a bivalent vaccine against *M. hyopneumoniae* and PCV-2 was evaluated in pigs vaccinated at three weeks of age, showing a greater ADWG during the finishing period [2,47,48,49,50,51] or from vaccination to the slaughter period [50,51]. Remarkably, no correlation between MDA and ADWG was observed in vaccinated animals, evoking that ADWG was independent of the MDA present at the time of vaccination as already observed in other studies [36,52], and indicating no evidence of interference of vaccine efficacy by MDA levels of the pigs from the tested herds.

A high mortality was detected in field trial B compared to the historical mortality in the farm, probably related to an outbreak of *Streptococcus suis* or *Glaesserella parasuis* infection, since gross lesions associated with these pathogens (fibrinous polyserositis, fibrinous pericarditis and/or polyarthritis) were observed in a high number of necropsied pigs. However, no significant effect of the vaccine on mortality was found in any of the studies, in agreement with some studies where the PCV-2-*M. hyopneumoniae* combined vaccine was administered in three-week-old pigs [2,47,49,50,52]. However, our and these mentioned results contrasted with other studies where a significantly lower mortality was observed in vaccinated animals [48,53] compared to non-vaccinated ones. It is noteworthy that the present field studies were designed with vaccinated and non-vaccinated commingled within the same pens, so, globally, vaccinated pig benefits could be worsened and non-vaccinated detriments could be ameliorated due to an overall increase of infectious pressure for vaccinated animals and a lower one for non-vaccinated ones [53].

Vaccination of pigs with one dose at three weeks of age with the trivalent vaccine reduced the IHC scorings in vaccinated animals significantly (PCV-2b pre-clinical and both clinical trials) or numerically (PCV-2a pre-clinical trial). Additionally, a significantly lower percentage of pigs with lymphoid lesions (when HR and LD were analysed together and when HR was analysed alone) was detected in the field trials. These results are in concordance with those observed in the split-dose vaccination at three days of age and three weeks later with the same trivalent vaccine used in this work [26]. Additionally, in the study of Park et al. [32], where a PCV-2-*M. hyopneumoniae* combined vaccine was administered at three weeks of age and a challenge three weeks later with PCV-2 and *M. hyopneumoniae* was performed, the reduction of the percentage of animals with lymphoid lesions and PCV-2 positive cells in their lymph nodes in vaccinated pigs compared to non-vaccinated ones was demonstrated.

Additionally, incidences of PCVD-SD and PCV-2-SI from both field studies (A and B) were numerically and statistically higher, respectively, in non-vaccinated groups compared to those of vaccinated ones, further indicating that vaccination reduces the clinical and subclinical impact of PCV-2 infection.

Vaccination generated a higher level of IgG antibodies after an experimental PCV-2a or PCV-2b challenge (in pre-clinical studies) or after a natural infection (in field studies), resulting in a faster humoral immune response upon infection. Such response paralleled with a reduction of PCV-2 loads in serum, faecal excretion, percentage of PCV-2 viraemic pigs (evaluated in clinical studies) and percentages of ever viraemic animals (except in the PCV-2b challenge pre-clinical study). The results agree with several studies under experimental and field conditions where piglets were injected with a combined PCV-2-*M. hyopneumoniae* vaccine or placebo at different ages (three days of age plus three weeks later, three or four weeks of age) [2,26,32,33,48,49,50,54] and PCV-2 viraemia and/or faecal excretion were significantly reduced in the vaccinated group compared to the placebo group.

Levels of MDA are very important for piglet immune response success upon vaccination [55] and the potential MDA interference on vaccine efficacy has not been yet demonstrated under field conditions [56], except in very particular situations with extremely high antibody values at vaccination [57,58]. In both field trials, a statistically significant negative correlation was detected between PCV-2 IgGs before vaccination and antibody values at seven weeks of age in all vaccinated animals, indicating a PCV-2-elicited antibody response of the vaccine dependent on MDA titres. These results indicate that lower PCV-2 S/P ratio levels should, ideally, ensure a seroconversion response after vaccination. However, it has been widely demonstrated that MDA do interfere with vaccine seroconversion [52,57,59,60,61], although not in all studies [26,36,40]. Importantly, such negative MDA effect on vaccine-elicited humoral immune response is not apparently related with a reduction of vaccine efficacy as observed in the present and other studies [50,56,60]. However, it is also evident that vaccine efficacy cannot be measured by vaccine seroconversion since not only humoral response, but also cell-mediated response, is involved in the protection against PCV-2 [2,50,57,60,62].

PCV-2 genotype co-infection (PCV-2a and PCV-2d) was found within the same farm in field trial B, a fact that has been described elsewhere [10,26,63,64]. In contrast, only PCV-2b was found in field trial A. The new trivalent vaccine assayed in the present studies contains cPCV-2a and cPCV-2b genotypes. However, several experimental works have shown cross-protection between the major genotypes worldwide (PCV-2a, PCV-2b and PCV-2d) [22,41,42,43,44,45] and a closer relation between PCV-2b and PCV-2d compared to PCV-2a and PCV-2d genotypes [20,23]. However, further studies would be necessary to corroborate this.

## 5. Conclusions

According to the results obtained globally in pre-clinical and field studies, a single immunisation at three weeks of age approximately with the novel PCV-2a/PCV-2b/*M. hyopneumoniae* vaccine was effective against PCV-2 infection (PCV-2a, PCV-2b or mixed PCV-a/PCV-2d) by reducing productive losses, viral load and shedding and histopathological lymphoid lesions.

## Figures and Tables

**Figure 1 vaccines-10-02108-f001:**
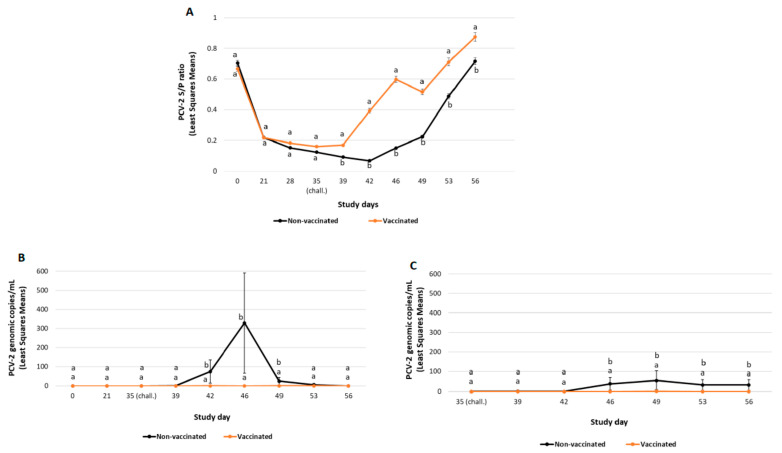
PCV-2a challenge study results: PCV-2 ELISA (mean S/P ratio ± SE) (panel **A**), PCV-2 viraemia load (mean PCV-2 DNA copies/mL ± SE) (panel **B**) and PCV-2 faecal shedding load (mean PCV-2 DNA copies/mL ± SE) (panel **C**). Different letters indicate significant differences among experimental groups (*p* ≤ 0.05).

**Figure 2 vaccines-10-02108-f002:**
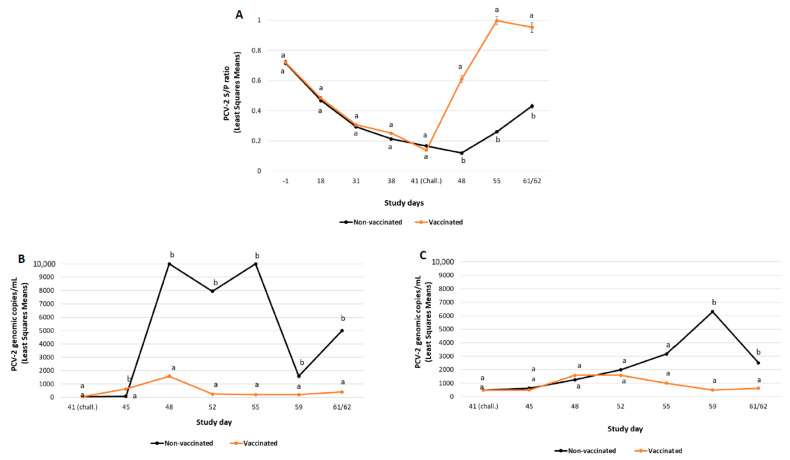
PCV-2b challenge study results: PCV-2 ELISA (mean S/P ratio ± SE) (panel **A**), PCV-2 viraemia load (mean log_10_ PCV-2 DNA copies/mL ± SE) (panel **B**) and PCV-2 faecal shedding load (mean log_10_ PCV-2 DNA copies/mL ± SE) (panel **C**). Different letters indicate significant differences among experimental groups (*p* ≤ 0.05).

**Figure 3 vaccines-10-02108-f003:**
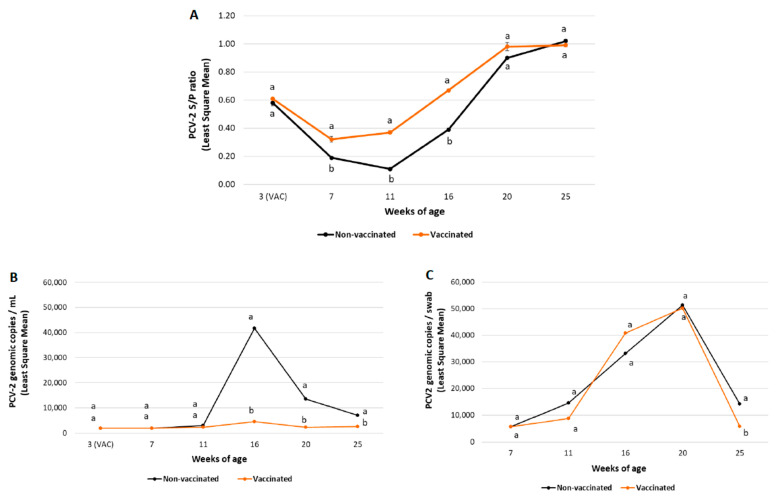
Field trial A results: PCV-2 IgG ELISA S/P results (mean ± SE) in serum samples (panel **A**), PCV-2 viraemia evolution (mean log_10_ genomic copies/mL ± SE) (panel **B**) and PCV-2 qPCR results (mean log_10_ genomic copies/swab ± SE) in faecal samples (panel **C**) at different timepoints. Different letters indicate significant differences among experimental groups (*p* ≤ 0.05).

**Figure 4 vaccines-10-02108-f004:**
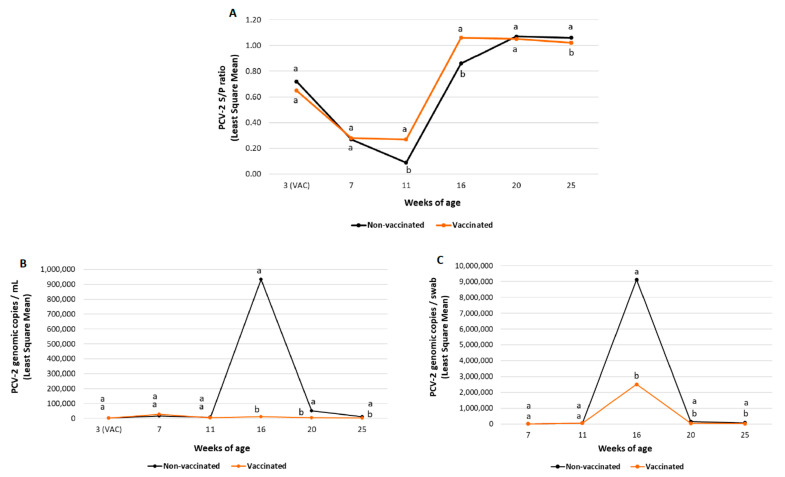
Field trial B results: PCV-2 IgG ELISA S/P results (mean ± SE) in serum samples at different timepoints (panel **A**), PCV-2 viraemia evolution (mean log_10_ genomic copies/mL ± SE) (panel **B**) and PCV-2 qPCR results (mean log_10_ genomic copies/swab ±SE) in faecal samples (panel **C**) at different timepoints. Different letters indicate significant differences among experimental groups (*p* ≤ 0.05).

**Figure 5 vaccines-10-02108-f005:**
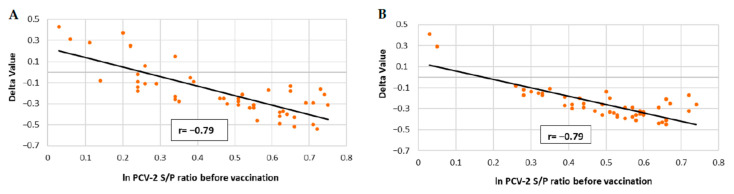
Linear regression and Pearson correlation coefficient between PCV-2 ELISA S/P ratios at vaccination and the increase of these titres until seven weeks of age approximately (Delta Value) in vaccinated piglets of field trial A (**A**) and field trial B (**B**).

**Table 1 vaccines-10-02108-t001:** Experimental study design and vaccination schedule of pre-clinical studies.

Experimental Groups	Pre-Clinical Studies									
	PCV-2a Challenge Experiment					PCV-2b Challenge Experiment				
	Mean S/P Ratio ± SE (PCV-2 MDA) at SD0	N	Vaccine or Placebo Administration	Challenge *	Necropsy	Mean S/P Ratio ± SE (PCV-2 MDA) at SD0	N	Vaccine or Placebo Administration	Challenge *	Necropsy
Non-vaccinated	0.71 ± 0.02 (seropositive)	24	SD0 (20–21 days old)	SD35	SD56	0.72 ± 0.01 (seropositive)	19	SD0 (18–23 days old)	SD41	SD61–62
Vaccinated	0.67 ± 0.02 (seropositive)	22				0.72 ± 0.01 (seropositive)	16			

SD: Study day; * Challenge was performed when the level of PCV-2 MDAs in the non-vaccinated group became low or undetectable (Geometric mean ELISA S/P ratio ≤ 0.2).

**Table 2 vaccines-10-02108-t002:** Experimental study design and vaccination schedule of clinical studies.

Field Trial	Farm	Treatment	Num. of Animals	Doses and Volume	Age at Vaccination
Field trial A	Farm A	Vaccinated	1013	2 mL	19–27 (ℜ = 24)
		Non-vaccinated	1011	2 mL	
Field trial B	Farm B	Vaccinated	1024	2 mL	18–24 (ℜ = 23)
		Non-vaccinated	1028	2 mL	

**Table 3 vaccines-10-02108-t003:** Proportion and percentage of ever PCV-2 viraemic or ever PCV-2 faecal shedding pigs in PCV-2a and PCV-2b challenge studies.

Group	PCV-2a Challenge Studies		PCV-2b Challenge Studies	
	Percentage of Ever Viraemic Pigs	Percentage of Ever Faecal Shedding Pigs	Percentage of Ever Viraemic Pigs	Percentage of Ever Faecal Shedding Pigs
Non-vaccinated	16/24 (66.6%) ^a^	14/24 (58.3%) ^a^	18/19 (94.7%) ^a^	18/19 (94.7%) ^a^
Vaccinated	1/22 (4.5%) ^b^	1/22 (4.5%) ^b^	13/16 (81.3) ^a^	10/16 (62.5) ^a^

Different letters indicate significant differences among experimental groups non-vaccinated and vaccinated (*p* ≤ 0.05).

**Table 4 vaccines-10-02108-t004:** Histopathology (histiocytic replacement (HR), lymphoid depletion (LD)) and immunohistochemistry (IHC) results (score > 0) in any of the four lymphoid tissues evaluated (mesenteric lymph node, inguinal superficial lymph node, tracheobronchial lymph node and tonsil) from the PCV-2a and PCV-2b challenge studies. Different letters indicate significant differences among experimental groups (*p* ≤ 0.05).

**Group**	**PCV-2a Challenge Studies**				**PCV-2b Challenge Studies**			
	**HR**	**LD**	**HR + LD**	**IHC**	**HR**	**LD**	**HR + LD**	**IHC**
Non-vaccinated	3/24 (12.5%) ^a^	4/24 (16.7%) ^a^	4/24 (16.7%) ^a^	2/24 (8.3%) ^a^	4/19 (21.1%) ^a^	1/19 (5.3%) ^a^	4/19 (21.1%) ^a^	5/19 (26.3%) ^a^
Vaccinated	0/22 (0.0%) ^a^	2/22 (9.1%) ^a^	2/22 (9.1%) ^a^	0/22 (0.0%) ^a^	1/16 (6.3) ^a^	0/16 (0.0) ^a^	1/16 (6.3) ^a^	0/16 (0.0) ^b^

**Table 5 vaccines-10-02108-t005:** Mean body weight (kg ± SE), average daily weight gain (ADWG, kg/day) and mortality for each field trial. Different letters indicate significant differences among experimental groups (*p* ≤ 0.05) for each field trial.

Study	Group	Body Weight (Kg ± SE)			ADWG (Kg/Day)			Mortality	
		3 WOA (vac)	16 WOA	24–27 WOA	3 WOA to 16 WOA	16 WOA to 24–27 WOA	3 WOA to 24–27 WOA	Each Treatment Group	Total
Field trial A	Vaccinated	5.9 ± 0.29 ^a^	51.1 ± 0.56 ^a^	100.7 ± 0.78 ^a^	0.49 ^a^	0.62 ^a^	0.78 ^a^	89/953 (9.3%) ^a^	181/1910 (9.5%)
	Non-vaccinated	5.9 ± 0.29 ^a^	50.3 ± 0.56 ^a^	99.1 ± 0.70 ^a^	0.48 ^a^	0.60 ^a^	0.76 ^a^	92/957 (9.6%) ^a^	
Field trial B	Vaccinated	5.7 ± 0.07 ^a^	45.6 ± 0.45 ^a^	105.0 ± 0.70 ^a^	0.43 ^a^	0.59 ^a^	0.76 ^a^	194/899 (21.6%) ^a^	402/1797 (22.4%)
	Non-vaccinated	5.7 ± 0.07 ^a^	44.2 ± 0.45 ^b^	99.6 ± 0.70 ^b^	0.42 ^b^	0.56 ^b^	0.71 ^b^	208/898 (23.2%) ^a^	

WOA: Weeks of age.

**Table 6 vaccines-10-02108-t006:** Proportion and percentage of PCV-2 qPCR-positive pigs (>3.3 log_10_ DNA copies/mL) at least in one sample point for each experimental group and field trial. Different letters indicate significant differences among experimental groups (*p* ≤ 0.05) for each field trial.

Study	Group	Proportion (%) of Pigs Detected Viraemic per Sampling Point						Total Proportion (%) of Ever Viraemic Pigs *
		3 WOA (vac)	7 WOA	11 WOA	16 WOA	20 WOA	25 WOA	
Field trial A	Vaccinated	0/50 (0.0%) ^a^	0/48 (0.0%) ^a^	3/47 (6.4%) ^a^	13/45 (28.9%) ^a^	6/47 (12.8%) ^a^	4/45 (8.9%) ^a^	22/45 (48.9%) ^a^
	Non-vaccinated	0/52 (0.0%) ^a^	0/51 (0.0%) ^a^	5/49 (10.2%) ^a^	30/46 (65.2%) ^b^	36/47 (76.6%) ^b^	22/46 (47.8%) ^b^	44/46 (95.7%) ^b^
Field trial B	Vaccinated	0/51 (0.0%) ^a^	28/48 (58.3%) ^a^	5/39 (12.8%) ^a^	26/58 (44.8%) ^a^	18/58 (31.0%) ^a^	12/57 (21.1%) ^a^	45/64 (70.3%) ^a^
	Non-vaccinated	0/51 (0.0%) ^a^	23/49 (46.9%) ^a^	8/39 (20.5%) ^a^	61/61 (100%) ^b^	57/59 (96.6%) ^b^	39/58 (67.2%) ^b^	65/65 (100%) ^b^

WOA: weeks of age. * Negative animals with a missing value in any of the timepoints were excluded from the analysis.

**Table 7 vaccines-10-02108-t007:** Proportion of animals with histopathology (histiocytic replacement (HR) and lymphoid depletion (LD)) and immunohistochemistry (IHC) results scores > 0 in at least one of the four lymphoid tissues evaluated (mesenteric lymph node, superficial inguinal lymph node, tracheobronchial lymph node and tonsil) corresponding to pigs which died or were euthanised during the study. Different letters indicate significant differences among experimental groups (*p* ≤ 0.05) within each field trial.

Study	Group	HR	LD	HR + LD	IHC
Field trial A	Vaccinated	3/111 (2.7%) ^a^	6/111 (5.4%) ^a^	7/111 (6.3%) ^a^	4/110 (3.6%) ^a^
	Non-vaccinated	12/116 (10.3%) ^b^	15/116 (12.9%) ^a^	19/116 (16.4%) ^b^	22/116 (19.0%) ^b^
Field trial B	Vaccinated	4/218 (1.8%) ^a^	30/218 (13.8%) ^a^	30/218 (13.8%) ^a^	9/228 (3.9%) ^a^
	Non-vaccinated	21/245 (8.6%) ^b^	39/245 (15.9%) ^a^	39/245 (15.9%) ^a^	43/253 (17.0%) ^b^

## Data Availability

Data are contained within the article.

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
