# Peer review of "Efficacy Studies against PCV-2 of a New Trivalent Vaccine including PCV-2a and PCV-2b Genotypes and *Mycoplasma hyopneumoniae* When Administered at 3 Weeks of Age"

_vaccines, 2022, doi:10.3390/vaccines10122108_

Round 1
Reviewer 1 Report
This study is well set. The manuscript is well prepared and is acceptable for publication. The authors should consider spelling out number less than 10.
Author Response
Response to Reviewer #1 Comments
This study is well set. The manuscript is well prepared and is acceptable for publication. The authors should consider spelling out number less than 10
Response: We really appreciate the positive reviewer comments. Numbers less than 10 have been written in words, except in those cases that corresponded to a serial (i.e., 7, 12 and 16 weeks of age).

Reviewer 2 Report
I would suggest the authors not abbreviating the terms vaccinated and un-vaccinated to V and UV, respectively, as I do not see any point in abbreviating single-word terms of this length.
With regard to the term "split-dose" - the authors have evaluated the formulation as a single-dose in the current study. It is unclear to me if the single dose used in the current study is half (i.e. split) dose used in the previous study. Or if the previous study used two doses, where each dose was equivalent to the dose used in the current study. I suspect the latter is true and would suggest to the authors that the term "split dose" does not apply. Please consider and review as/if necessary.
Suggest adding "single dose" to the keywords.
Numbers less than 10 should be written in words - this convention is inconsistently applied in the manuscript.
Author Response
Response to Reviewer #2 Comments
I would suggest the authors not abbreviating the terms vaccinated and un-vaccinated to V and UV, respectively, as I do not see any point in abbreviating single-word terms of this length.
Response: We really appreciate the reviewer suggestion. Terms vaccinated and un-vaccinated have been modified.
With regard to the term "split-dose" - the authors have evaluated the formulation as a single-dose in the current study. It is unclear to me if the single dose used in the current study is half (i.e. split) dose used in the previous study. Or if the previous study used two doses, where each dose was equivalent to the dose used in the current study. I suspect the latter is true and would suggest to the authors that the term "split dose" does not apply. Please consider and review as/if necessary.
Response: We appreciate the observation of the reviewer. The split-dose used in the previous study was half dose each time (1 mL+1 mL) of the dose used in the current study (2 mL). The “split-dose” definition is the division of the daily dose into two or more administrations. Taking onto account that one full dose is 2 mL, and that in the previous study it was administrated 1 mL+1 mL, we consider that it could be indicated as “split-dose”.
Suggest adding "single dose" to the keywords.
Response: “Single dose” has been added in the keywords (line 46 of the revised version of the manuscript).
Numbers less than 10 should be written in words - this convention is inconsistently applied in the manuscript.
Response: In agreement with the comments of the reviewer, numbers less than 10 have been written in words, except in those cases that corresponded to a serial (i.e., 7, 12 and 16 weeks of age).

Reviewer 3 Report
This study aimed to evaluate the efficacy of a new trivalent vaccine containing inactivated Porcine Circovirus 1-2a and 1-2b chimeras and a Mycoplasma hyopneumoniae bacterin administered to pigs around 3 weeks of age. The result is useful in guiding the field use of trivalent vaccine. Minor revision is needed before publication.
1. add the result and conclusion in the Abstract section
2. Figure 1C, the curve in the figure 1C is too low. Change the y-axis value below 100.
3. Why do you use 3-weeks-age pig in this study? Did the maternal antibody affect the results?
Author Response
Response to Reviewer #3 Comments
This study aimed to evaluate the efficacy of a new trivalent vaccine containing inactivated Porcine Circovirus 1-2a and 1-2b chimeras and a Mycoplasma hyopneumoniae bacterin administered to pigs around 3 weeks of age. The result is useful in guiding the field use of trivalent vaccine. Minor revision is needed before publication.
Response: We really appreciate the positive reviewer comments and we tried to address all her/his suggestions as follows.
- add the result and conclusion in the Abstract section
Response: We appreciate the observation of the reviewer. However, the results and conclusions were already included in the Abstract (lines 34-43 of the revised version of the manuscript).
- Figure 1C, the curve in the figure 1C is too low. Change the y-axis value below 100.
Response: We appreciate the reviewer suggestion. The y-axis value has been fixed in 600 as in the y-axis value of figure 1B (PCV-2 viraemia load), to visually facilitate the comparison of viral loads between the two graphs. By using two different scales in the y-axes in the two different figures would cause a misleading when comparing them. Moreover, Figures B and C from each study have been located at the same line to improve the comparison.
- Why do you use 3-weeks-age pig in this study? Did the maternal antibody affect the results?
Response: It was chosen 3-weeks-age pigs to evaluate the efficacy the trivalent vaccine because the weaning period is the most common timing of PCV-2 vaccination in the production industry as indicated in lines 660-663 of the revised version of the manuscript.
The main objective of the study was to assess the efficacy of the product, so although a negative MDA effect on humoral immune response to the vaccine was observed in the present study and it has been described elsewhere (Segalés, 2015), this was not apparently related with a reduction of vaccine efficacy as indicated in lines 746-754 of the revised version of the manuscript.

Reviewer 4 Report
The research presented here addresses a topic of great importance due to the economic burden of PCV-2 on the pork industry. The increasing prevalence of PCV-2d and other strains, likely driven by existing vaccines based primarily on PCV-2a, has reduced the effecacy of earlier vaccines. Therefore, development of new vaccines is required, their effectiveness tested, and optimal dosing regimens identified.
Major strengths of this work include:
· Thoughtful commentary in the introduction and discussion sections places these findings in the proper context of current PCV2 vaccine research.
· The authors present an impactful weight of evidence from combined pre-clinical and clinical studies with large sample sizes, convincing timecourse data, and appropriate statistical analyses.
· Genotyping of the PCV-2 strains circulating in the field tests is an important addition, as the antigenic crossreactivity of different strains is not thought to be uniform, and the spectrum of protection elicited by vaccination is a key subject for investigation.
As a major stated goal of this work is to determine the adequacy of a single dose vaccination regimen, it would be good to mention in discussion the relative efficacy of the single dose reported here versus the efficacy previously demonstrated with a split dosing regimen as published by these authors earlier this year.
It would be interesting in future studies to challenge with PCV-2d or other newer strains.
Author Response
Response to Reviewer #4 Comments
The research presented here addresses a topic of great importance due to the economic burden of PCV-2 on the pork industry. The increasing prevalence of PCV-2d and other strains, likely driven by existing vaccines based primarily on PCV-2a, has reduced the effecacy of earlier vaccines. Therefore, development of new vaccines is required, their effectiveness tested, and optimal dosing regimens identified.
Major strengths of this work include:
- Thoughtful commentary in the introduction and discussion sections places these findings in the proper context of current PCV2 vaccine research.
- The authors present an impactful weight of evidence from combined pre-clinical and clinical studies with large sample sizes, convincing timecourse data, and appropriate statistical analyses.
- Genotyping of the PCV-2 strains circulating in the field tests is an important addition, as the antigenic crossreactivity of different strains is not thought to be uniform, and the spectrum of protection elicited by vaccination is a key subject for investigation.
As a major stated goal of this work is to determine the adequacy of a single dose vaccination regimen, it would be good to mention in discussion the relative efficacy of the single dose reported here versus the efficacy previously demonstrated with a split dosing regimen as published by these authors earlier this year.
Response: We appreciate the observation of the reviewer. Considering both vaccination regimens interesting for the porcine industry, comparison of both regimes was not the objective of these studies. Both may help preventing PCV-2 infection depending on the infection dynamics of the farm as mentioned in lines 663-671 of the revised version of the manuscript.
It would be interesting in future studies to challenge with PCV-2d or other newer strains.
Response: We really appreciate the reviewer suggestion. With the objective of updating a PCV-2a based monovalent vaccine (Suvaxyn Circo+MH RTU) to provide a broader immunological coverage, PCV-2b was selected since the combination of PCV-2a and PCV-2b could provide cross-protection for PCV-2d challenge, based on T cell epitope content comparison (EpiCC) analysis as demonstrated in Bandrick et al., 2020. In Bandrick et al. it was shown that a bivalent PCV-2 vaccines have greater T cell epitope overlap with field strains than monovalent PCV-2 vaccines. Moreover, a closer epitopic relationship between PCV-2b and PCV-2d than between PCV-2a and PCV-2d genotypes has been demonstrated. However, as suggested by the reviewer, it would be also interesting include a PCV-2d challenge since it is the predominant PCV-2 genotype in Europe currently.
